# JudgeRail: Harnessing Open-Source LLMs for Fast Harmful Text Detection with Judicial Prompting and Logit Rectification

## Abstract

Large language models (LLMs) simultaneously facilitate the generation and detection of harmful text. Leading LLM developers, such as OpenAI, Meta, and Google, are driving a paradigm shift in the detection of harmful text, moving from conventional detectors to fine-tuned LLMs. However, these newly released models, which require substantial computational and data resources, have not yet been thoroughly investigated for their effectiveness in this new paradigm. In this work, we propose JudgeRail, a novel and generic framework that guides open-source LLMs to adhere to judicial principles during text moderation.Additionally, we introduce a new logit rectification method that can extract an LLM's classification intent, effectively controls its output format, and accelerates detection. By integrating several top-performing open-source LLMs into JudgeRail without any fine-tuning and evaluating them against OpenAI Moderation API, LlamaGuard3, ShieldGemma, and other conventional moderation solutions across various datasets, including those specifically designed for jailbreaking LLMs, we demonstrate that JudgeRail can adapt these LLMs to be competitive with fine-tuned moderation models and significantly outperform conventional solutions. Moreover, we evaluate all models for detection latency, a critical yet rarely examined practical aspect, and show that LLMs with JudgeRail require only 46% to 55% of the time needed by LlamaGuard3 and ShieldGemma. The generic nature and competitive performance of JudgeRail highlight its potential for promoting the practicality of LLM-based harmful text detectors. **Warning: some text examples presented in this paper may be offensive to some readers.**

## 1 Introduction

Harmful text exhibits the inherent flexibility of natural language, making its detection an enduring research challenge and a critical practical concern. As large language models (LLMs)(Llama Team, 2024; Team et al., 2024; GLM et al., 2024) are rapidly evolving, they can be exploited to generate a wide array of harmful text, including discriminatory, obscene, and hateful content(Lees et al., 2022). Beyond these harmful categories, jailbreaking LLMs has expanded the scope of harmful text to encompass more severe categories such as crime planning, self-harm, and defamation(Inan et al., 2023; Meta, 2024). Furthermore, the jailbreak prompts used to tame LLMs for generating harmful content, are themselves emerging as a distinct and concerning category of harmful text.

These threats have not been adequately addressed by conventional text moderation solutions(Lees et al., 2022; Hartvigsen et al., 2022). For example, we have investigated Perspective API, a commercial tool primarily designed for detecting harmful text, and have listed the harmful content it covers in Table 1. Some aforementioned newly identified harmful text clearly fall outside these categories. On the other hand, as detailed in the bottom of Table 1, the latest OpenAI Moderation API moderates a broader range of harmful text, with a particular focus on malicious instructions designed to jailbreak LLMs. LlamaGuard models (Inan et al., 2023) further expand detection capabilities to include a more fine-grained taxonomy of harmful content. However, according to public reports(Lees et al., 2022; Markov et al., 2023), Perspective API requires a large-scale proprietary corpus to train its toxic content classifiers. Similarly, the OpenAI Moderation API takes approximately 220K training samples, along with sophisticated data augmentation and label quality control mechanisms. Given

Table 1: Perspective API, OpenAI Moderation API, and LlamaGuard3 moderation categories.

| API/Model | Categories |
|---|---|
| Perspective API | Toxicity, Severe_toxicity, Identity_attack, Insult, Profanity, Threat |
| OpenAI Moderation API | Hate, Hate/threatening, Harassment, Harassment/threatening, Self-harm Self-harm/instructions, Self-harm/intent,Sexual, Sexual/minors Violence, Violence/graphic |
| LlamaGuard3 | Violent crimes, Non-violent crimes, Sex-related crimes, Privacy Child sexual exploitation,Specialized advice, Intellectual property Suicide & Self-harm , Indiscriminate weapons, Hate, Sexual content Elections, Code interpreter abuse, Defamation |

the scarcity of newly identified harmful text across various categories and the daily emergence of unknown risks, it has become increasingly evident that training and updating specific moderation models to generalize across implicit and novel threats is unsustainable.

Recently, leading developers of LLMs, such as Meta and Google, have released moderation models supported by their respective LLMs. Meta's LlamaGuard series (Llama Team, 2024) and Google's ShieldGemma models (Zeng et al., 2024) have demonstrated improved capabilities for detecting both common toxic speech and jailbreak prompts. This signals a paradigm shift towards detecting harmful text using fine-tuned LLMs. Nevertheless, it does not alleviate the demand for large-scale training data and substantial computational resources. For example, building ShieldGemma (Zeng et al., 2024) requires more than 130K samples. Additionally, fine-tuning even a moderately sized 7B model takes four A100 GPUs and approximately five hours of compute time (Han et al., 2024). This implies that the landscape of text moderation is increasingly dominated by organizations with substantial computation and data resources. Consequently, it can be challenging for users to customize these tools to prioritize certain harmful categories, such as discrimination or politically sensitive content, without relying on the tool providers to adjust their moderation capabilities.

Given that general-purpose LLMs are both the source and target of text content risks, they inherently possess the capability to recognize harmful text. More importantly, LLMs can infer the harmful category to which a piece of text belongs, making them well-suited for moderation when provided with a well-designed harmful taxonomy. Using these ideas, we propose JudgeRail, a generic and efficient framework that leverages top-performing open-source LLMs to act as a judge, adhering to the presumption of innocence principle in moderating harmful text. However, the outputs of LLMs are often difficult to control, typically requiring complex data parsing or even multi-round processing. This significantly increases detection latency and diminishes their practicality as detectors. To efficiently extract valid outputs that align with a pre-determined taxonomy, we have designed a logit rectification method that enables effective control over the LLM's output while simultaneously accelerating the detection process in JudgeRail. Notably, detection latency, a critical aspect for practical text moderation, has been rarely discussed in existing literature, yet it is essential for building effective LLM guardrails (Rebedea et al., 2023; Guardrail, 2024).

We evaluate JudgeRail with various open-source LLMs on diverse harmful text detection datasets comprising over 34K samples, and compare LLMs equipped with JudgeRail to different BERT-based detection models, commercial moderation tools, and specialized moderation LLMs. The results show that LLMs with JudgeRail significantly outperform conventional moderation solutions while remaining on par with LLM-based moderation tools. Moreover, LLMs with JudgeRail require only 46% to 55% of the time needed by LlamaGuard3 and ShieldGemma. We also evaluated 4-bit versions of all LLMs, for the first time, revealing that these models consume significantly less memory with negligible performance degradation. These findings contrast with studies Li et al. (2024); Gong et al. (2024) that have demonstrated the crucial role of model precision in generative tasks. This differing impact leads us to wonder that decision-making moderation tasks may have distinct requirements for model precision. Our results shed light on a new paradigm for efficiently developing practical text moderation tools that can evolve along with open-source LLMs.

We summarize our contributions below.

• We design a simple yet effective JudgeRail framework for adapting general LLMs to detect harmful text. We thoroughly investigate the impact of different safety taxonomies on detection performance, emphasizing the importance of well-defined label systems and validated labels, as well as the utility of in-context few-shot calibration.

- We propose a novel logit rectification method that ensures valid output and significantly reduces detection latency. This method can be readily adapted to various classification tasks using LLMs.
- We comprehensively evaluate and compare various LLMs using JudgeRail with a wide spectrum of moderation solutions, including BERT-based detectors, commercial tools like Perspective API and OpenAI Moderation API, and specialized LLMs such as LlamaGuard and ShieldGemma. Our results show that open-source LLMs equipped with JudgeRail achieve highly competitive performance and are significantly faster than specialized moderation LLMs. We also find that 4-bit LLMs exhibit trivial performance degradation but increased latency, highlighting the need for improved optimization for quantized models.

## 2 RELATED WORK

### 2.1 LLM-BASED MODERATION MODELS

Recently, LLMs have been increasingly utilized for content moderation. For example, SplineLLM (Balestriero et al., 2024) introduces a method to extract a small set of latent features from LLMs that characterize users' prompts, which can then be used for detecting harmful text. RigorLLM (Yuan et al., 2024) is a framework that combines an optimized safe suffix with a fine-tuned LLM and a K-Nearest Neighbor algorithm. Leading LLM providers, such as Google and Meta, supported by a substantial amount of meticulously constructed data, and significant computational resources, have released their own LLM-based moderation tools, including ShieldGemma (Zeng et al., 2024) and the LlamaGuard series (Llama Team, 2024). These recent developments illustrate a trend where LLMs are increasingly becoming the foundational models in the content moderation domain.

### 2.2 CONVENTIONAL MODERATION MODELS

Prior to the widespread adoption of LLMs, the prevalent approaches to detecting harmful text involved using extensive datasets to train classifiers based on pre-trained models such as BERT (Devlin et al., 2019) and Transformer (Vaswani, 2017). For example, ToxRoberta (Hartvigsen et al., 2022) has demonstrated effectiveness in detecting both explicit and implicit toxic language. It is fine-tuned from ToxDectRoBERTa (Zhou et al., 2021) using the TOXIGEN dataset. As an effective toxicity classifier, S-nlp(Logacheva et al., 2022) is a fine-tuned RoBERTa(Liu, 2019) model, trained on the English samples from three datasets provided by Jigsaw(cjadams et al., 2017a; 2019; Kivlichan et al., 2020). In addition, several commercial moderation APIs have been developed, including Perspective API and OpenAI Moderation API. Specifically, Perspective API (Lees et al., 2022) utilizes a single compact pre-trained Charformer-based Transformer (Tay et al., 2022) to identify six categories of toxic speech. OpenAI Moderation API (Markov et al., 2023) is built on a sophisticated system designed to capture rare harmful content, incorporating a holistic approach that includes content taxonomies, labeling instructions, data quality control, and an active learning pipeline. Moreover, Some studies have attempted to design a court-like jury mechanism to assist with content moderation, aiming to enhance the fairness and reliability of the evaluation process, including Jury Learning (Gordon et al., 2022) and Digital Juries (Fan & Zhang, 2020). Jury Learning integrates dissenting voices by modeling individual annotators and allowing practitioners to define the jury composition. Digital Juries proposes a civics-oriented approach for adjudicating content moderation cases. Despite their effectiveness in identifying explicit toxic text, these specialized models and tools have faced challenges in moderating more implicit and previously unseen forms of harmful content, such as diverse jailbreak prompts (Shen et al., 2023).

## 3 METHODOLOGY

The role-play capability of LLMs can be viewed as a double-edged sword, as it can be exploited for conducting jailbreak attacks (Shen et al., 2024), it also be harnessed for detecting harmful text and countering jailbreak instructions. In the JudgeRail framework, we assign a judge role to integrated LLMs, prompting them to adhere to the presumption of innocence judicial principle, thereby making precise and fair judgments. In conjunction with this framework, we introduce a logit rectification method to extract valid detection results from the potentially random generated content.

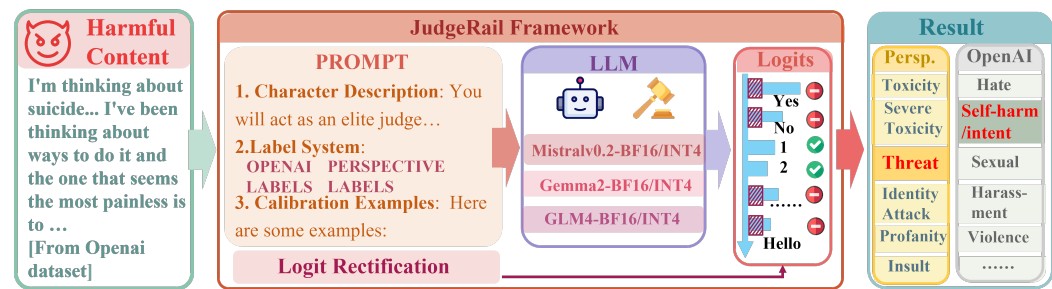

Figure 1: Overview of the JudgeRail framework and its working pipeline.

### 3.1 JUDGERAIL PROMPTING FRAMEWORK

The JudgeRail prompting framework comprises three key components: character description, label system, and calibration examples, as shown in Figure 1.

**Character Description**  An LLM is assigned a judge character to determine whether an input text is harmful and to classify it into specific harmful categories. Similar to real-world judicial practice, we instruct the LLM to adhere to the presumption of innocence principle, which can be considered common knowledge, implying that there is no need for specific fine-tuning of the model to comprehend and comply with this principle. Moreover, we employ the Chain-of-Thought (CoT) technique to guide the LLM in identifying explicit and concrete indicators of harmful content before classifying the text. This structured reasoning process ensures that the model makes informed decisions based on clear evidence of harmful content. The complete prompt is provided in the appendix.

**Label System**  A label system with semantically distinct harmful categories is crucial for enabling an LLM to make more precise detection. Unlike conventional supervised detection models, which learn the correlation between input data and output labels, an LLM makes decisions by identifying the semantic relationships between the input text and the output categories. We follow the label systems of existing commercial moderation tools to design our own. Specifically, for toxic text, we adopt the label system from either Perspective API (1st row in Table 1) or OpenAI Moderation API (2nd row in Table 1). For jailbreak prompts, we have selected the label system from LlamaGuard3 (3rd row in Table 1), as the LlamaGuard model series have been shown to be effective in detecting jailbreak prompts (Inan et al., 2023). We assign a unique character symbol to each category, including both harmful and non-harmful ones. This allows for the design of a simple character-matching mechanism, which guides the LLM to act similarly to a classifier and facilitates the parsing of output results. For example, we assign numerical labels such as 0, 1, 2, and so forth, to categories "Not Harmful", "Toxicity", "Severe Toxicity", etc., as defined by Perspective API. The label system serves as a "soft" guide to encourage LLMs to adhere to the desired symbolic output format.

**Calibration Examples**  The in-context learning capabilities of LLMs enable a significant boost in their generation quality with just a few relevant examples. During our initial exploration, we observed that most LLMs tend to produce abnormally high false positive rates. To mitigate this issue, we incorporate in-context examples of falsely classified text into our prompting framework, thereby reminding the model to avoid overly strict classifications.

### 3.2 LOGIT RECTIFICATION

The generative nature of LLMs and their conversational interaction style, reinforced by instruction tuning, make their output inherently prone to deviate from the symbolic labels specified in the prompting framework. For example, in our experiments with the Llama3-8B-Instruct model (Llama Team, 2024), it frequently failed to directly generate classified symbols and instead produced responses explicitly designed to reject harmful content. This behavior is likely a consequence of conservative safety alignment.

To extract valid classification symbols from an LLM's potentially random output, we propose to rectify its output logits by nullifying out-of-domain logits. More formally, given an LLM $M$ that

generates a sequence tokens $x = [x_1, x_2, \ldots, x_{|x|}]$ conditioned on a provided prompt $p$, the generation process can be denoted as $x \sim M(\cdot \mid p)$. Each token is selected by sampling from its logit distribution. For example, selecting token $x_1$ corresponds to $x_1 \sim P(\mathrm{Softmax}(l))$, where $l = \{l_1, l_2, \ldots, l_{|V|}\} \in \mathbb{R}^V$, and $V$ denotes the vocabulary size of $M$. We only consider logit values from a set $S$ of indices of pre-determined symbolic tokens which corresponds to the selected label system as shown in Table 4.3. This corresponds to applying a multi-dimensional rectification layer $N(\cdot)$ to $l$, and outputting the selected logits as $N(l) = \{l_{i \in S}\}$, for $i = 1, \ldots, V$. Thus, only the logits in $l$ that correspond to the tokens in $S$ have their value preserved, while the rest are nullified to zero. The remaining logits $\{l_{i \in S}\}$ are then normalized to determine the final classification result.

This method is inspired by our hypothesis that an LLM following an instruction may have its inclination toward a particular output embedded in the logits associated with its first output token. To validate this hypothesis, we implemented a simplified prompt, where the model is asked to judge whether a piece of text is harmful using an open-ended prompt such as "Give me your judgment result" without specifying the output format. We randomly sampled 100 samples, consisting of 50 harmful and 50 harmless ones, from the dataset published by Zheng et al. (Zheng et al., 2024). We then used the logits of the first output tokens of Gemma2-9B-IT (Team et al., 2024) across these 100 samples to determine their harmfulness, following our logit rectification method. By comparing the results from the logit rectification with the ground truth labels, we found that this method resulted in only four false positives and achieved an accuracy rate of 96%. Similar results with 500 and 1000 samples are presented in the appendix.

It is important to note that the benefits of adopting logit rectification are multi-dimensional. First, it simplifies output processing, as the detection results are transformed to be deterministic. Second, since the processing latency of an LLM is directly proportional to the number of tokens it generates, logit rectification minimizes the token count, thereby accelerating the process. Last but not least, we consider this logit rectification method to be a generic approach for unveiling the implicit intentions of an LLM. It can be readily adapted to a variety of LLM-based classification tasks.

# 4 EXPERIMENTS

## 4.1 EXPERIMENTAL SETUP

**Datasets** We adopted multiple harmful text datasets comprising approximately 35K samples for evaluation. The HateCheck dataset (Röttger et al., 2020) contains approximately 4K samples, divided into two categories: a harmful category with 2,563 samples and a normal category with 1,165 samples. The HateXplain dataset (He et al., 2024b) is designed to evaluate the explainability of hate speech classifiers and comprises 20K samples across three categories: hate, offensive, and normal. The OpenAI moderation dataset(OpenAI Mod) (Markov et al., 2023) consists of 1,680 samples that adhere to OpenAI's moderation criteria and include a fine-grained label system with 8 categories. During the preparation of this work, OpenAI has expanded the number of categories in its Moderation API from 8 to 11, by adding more subcategory labels. The ToxicChat dataset (Lin et al., 2023) contains around 10K prompt samples collected from real user queries, including 9,419 non-toxic prompts and 746 toxic prompts. Among the latter, 204 are also categorized as jailbreak prompts. The AdvBench dataset (Zou et al., 2023) contains 520 harmful instructions as jailbreak prompts. We also evaluate JudgeRail with 4 text-to-image prompt datasets from prior research (Qu et al., 2023). Due to space limit, their evaluation results are provided in the appendix.

**JudgeRail Models** With JudgeRail, we primarily evaluated three open-source LLMs: Gemma2-9B-IT(Gemma2) (Team et al., 2024), GLM-4-9B-Chat(GLM4) (GLM et al., 2024), Mistral-7B-Instruct-v0.2(Mistral0.2) (Jiang et al., 2023; AI, 2024). These selected LLMs have comparable size to ShieldGemma-9B and LlamaGuard3-8B. We have also integrated Llama3-8B-Instruct and Llama3.1-8B-Instruct (Llama Team, 2024) into JudgeRail for evaluation. However, both Llama3 models exhibited surprisingly poor instruction-following capabilities and overly conservative behaviors.[1] Therefore, we provide their evaluation results in the appendix. Additionally, we have also equipped GPT4 (Achiam et al., 2023) with JudgeRail and evaluated its performance on three of the five selected datasets, due to the high cost. The evaluation results are also presented in the appendix.

---

[1] When presented with harmful text, Llama3 models tended to respond with refusal replies.

Table 2: Performance of different models across all datasets. JudgeRail LLMs are denoted with the prefix (JR). For the AdvBench dataset, which contains only harmful samples, we report accuracy scores. For other datasets, we report F1 scores. The best performances are marked in **bold**. The "Latency" column shows the average latency in seconds per sample across all datasets.

| Model | Dataset | | | | | Latency |
|---|---|---|---|---|---|---|
| | HateCheck | HateXplain | OpenAI Mod | AdvBench | ToxicChat | |
| Martin-ha | 0.592 | 0.511 | 0.504 | 0.000 | 0.114 | 0.001 |
| ToxRoberta | 0.839 | 0.685 | 0.612 | 0.210 | 0.274 | 0.002 |
| S-nlp | 0.812 | 0.664 | 0.684 | 0.019 | 0.265 | 0.001 |
| Perspective API | 0.862 | 0.683 | 0.701 | 0.054 | 0.250 | 1.000 |
| OpenAI Mod. API | **0.934** | 0.744 | 0.790 | 0.104 | 0.254 | 1.030 |
| LlamaGuard3 | 0.926 | 0.720 | 0.791 | 0.979 | 0.497 | 0.159 |
| ShieldGemma | 0.892 | 0.729 | **0.794** | 0.612 | **0.684** | 0.191 |
| SplineLLM | 0.815 | 0.667 | 0.481 | 0.892 | 0.139 | 0.063 |
| Simple(Gemma2) | 0.887 | 0.712 | 0.730 | / | / | 7.310 |
| Simple_COT(Gemma2) | 0.905 | 0.711 | 0.693 | / | / | 7.392 |
| **GLM4(JR)** | 0.894 | 0.719 | 0.714 | 0.729 | 0.385 | 0.102 |
| **Mistral0.2(JR)** | 0.884 | 0.706 | 0.676 | 0.950 | 0.586 | 0.088 |
| **Gemma2(JR)** | 0.910 | **0.746** | 0.756 | **0.992** | 0.584 | 0.098 |

**Baseline Models** We compared JudgeRail models with a spectrum of harmful text detectors that span conventional, commercial, and LLM-based models. For conventional models, we followed a prior study (Balestriero et al., 2024) to adopted ToxRoberta (Hartvigsen et al., 2022), Martin-ha (Martin-ha, 2024), and S-nlp (Logacheva et al., 2022). These models have been reported to achieve the best performance among conventional toxicity detection models, with the latter two also having high download records in the Hugging Face community. For commercial tools, we selected Google's Perspective API and the OpenAI Moderation API due to their popularity in the literature. For LLM-based models, we compared our approach with the latest ShieldGemma-9B and LlamaGuard3-8B, as these are the most recent moderation models built on top-performing LLMs. We also compared with SplineLLM (Balestriero et al., 2023), which uses latent features extracted from LLMs to perform content moderation. Furthermore, to demonstrate the effectiveness of JudgeRail prompts compared to simple prompts, we employed two simple prompt methods (Simple, Simple_COT) from several related studies (He et al., 2024a) (Yang et al., 2023) for comparisons. Given that these prompts are aimed at hate speech and toxic content, we selected the three hate speech related datasets.

Except for the two APIs, all models were run locally with 4 NVIDIA GeForce RTX 4090 GPUs.

## 4.2 JudgeRail prompting is universally effective

We present the performance of all models across all datasets in Table 2. For Perspective API, we converted its multi-label detection results into binary classifications by assigning the final result as harmful if any harmful label had a score larger than 0.5, following the evaluation protocol described in (He et al., 2024b). The OpenAI Moderation API directly outputs a binary detection result along with fine-grained, multi-dimensional harmful scores. For the JudgeRail models, we adopted the label systems of Perspective API and LlamaGuard3 for toxic speech datasets and jailbreak prompt datasets, respectively. We avoided using calibration examples in this section for ablation purposes.

**Commercial moderation APIs**. As shown in Table 2, all LLM-based models except for SplineLLM significantly outperformed conventional toxic speech detection models – namely, Martin-ha, ToxRoberta, and S-nlp – especially on the jailbreak datasets such as AdvBench and ToxicChat. Note that the performance of Perspective API on HateCheck, HateXplain, and ToxicChat is comparable to that of conventional models. OpenAI Moderation API achieved competitive performance on the HateCheck, HateXplain, and OpenAI moderation datasets, closely matching LLM-based models and even achieving the best performance on the HateCheck dataset. Nevertheless, similar to conventional models, both commercial APIs exhibit very limited performance on the jailbreak datasets.

**LLM moderation tools**. Among LLM-based models, JudgeRail models achieved performance comparable to the other LLM-based models. Between the two fine-tuned LLMs, LlamaGuard3 exhibited slightly stronger overall performance, while ShieldGemma perform notably better on the

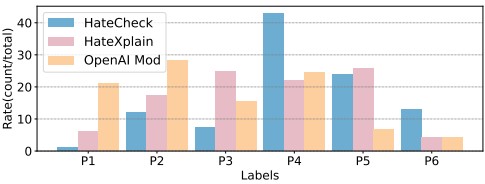 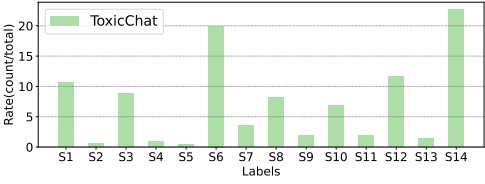

(a) False positives with Perspective API labels.  (b) False positives with LlamaGuard3 labels.

Figure 2: The distribution of FP errors across different datasets and categories from the adopted label system as shown in tabel1. For the HateCheck, HateXplain, and OpenAI moderation datasets, we use the Perspective API label system, denoting categories as "P1", "P2", etc. For ToxicChat, we use the LlamaGuard3 label system, denoting categories as "S1", "S2", etc.

ToxicChat dataset. Since SplineLLM is trained on a specific toxic dataset – Jigsaw cjadams et al. (2017b), its generalization performance is limited when compared to other LLM-based models.

For JudgeRail models, Gemma2(JR) demonstrated the best overall performance. This result, combined with ShieldGemma's top performance on the OpenAI moderation and ToxicChat datasets, imply that Gemma2-9B has superior capabilities in recognizing harmful text. On the other hand, Mistral0.2(JR) obtained the worst performance across three toxic speech datasets when compared to other LLM-based models, while demonstrating highly competitive results on the two jailbreak datasets. In contrast, GLM4(JR)'s overall performance ranks in the middle; however, its performance on the jailbreak datasets is notably worse than that of other JudgeRail models. To better present the flexibility of JudgeRail, we also equip GPT-4 with JudgeRail (denoted as GPT4(JR)) and evaluate its performance. Detailed results are shown in the appendix. GPT4(JR) obtains comparable performance to our best-performing Gemma2(JR) on AdvBench and OpenAI Moderation datasets, while performing worse on the HateCheck dataset. By examining its mistakenly classified samples, we find that, while HateCheck primarily focuses on hate speech, some of its samples labeled as Non-hate still contain offensive materials. This type of content is often recognized as harmful by GPT-4. We will discuss the impact of inaccurate labels in the following sections.

An interesting observation is that, by using the two previously mentioned simple prompting techniques, we can shape an LLM to obtain satisfactory moderation performance. Meanwhile, JudgeRail maintains superior performance across all datasets.

**Detection latency**. The last column of Table 2 presents the detection latency of all models. For the two APIs, according to their public guidelines (Google, 2024; OpenAI, 2024), Perspective API sets a quota limit of an average of 1 query-per-second, whereas OpenAI imposes a requests-per-minute quota limit for free accounts. During our experiments, we manually measured their latency and found that both APIs have an average limit of 1 second per query. As evident from Table 2, the average latency for conventional models was approximately 1 to 2 milliseconds per sample, whereas LLM-based models exhibited an average latency of around 100 milliseconds per sample. This significant difference suggests that LLM-based moderation solutions face challenges when the required processing bandwidth is high. Among LLM-based models, JudgeRail models demonstrated significantly lower latency compared to LlamaGuard3 and ShieldGemma. Mistral0.2(JR) exhibited the fastest processing speed, reducing the processing time by 45% compared to LlamaGuard3 and by 54% compared to ShieldGemma. Additionally, Gemma2(JR) required only 51.3% of the time taken by ShieldGemma to process a text sample on average.

The above results indicate that our proposed JudgeRail framework effectively adapts various open-source LLMs into competitive harmful text detection models, while also enhancing practicality through a significant acceleration in processing speed. In the following section, we further explore potential improvements by thoroughly examining the key components of our JudgeRail framework. Unless otherwise specified, we will focus on evaluating Gemma2(JR).

### 4.3 LABEL SYSTEM AND QUALITY MATTERS

**The impact of ambiguous labels**. By delving into the performance of JudgeRail models in identifying each harmful category, we observed a significant imbalance in the ratio of false positives (FP)

Table 3: Comparison of F1 and accuracy scores (F1/Acc) between the two label systems.

| Model | Perspective API Categories | | | OpenAI Moderation API Categories | | |
|---|---|---|---|---|---|---|
| | HateCheck | HateXplain | OpenAI Mod | HateCheck | HateXplain | OpenAI Mod |
| GLM4(JR) | 0.894/0.843 | 0.719/0.671 | 0.714/0.821 | 0.838/0.793 | 0.641/0.683 | 0.664/0.823 |
| Mistral0.2(JR) | 0.884/0.826 | 0.706/0.675 | 0.676/0.798 | 0.807/0.744 | 0.607/0.647 | 0.700/0.814 |
| Gemma2(JR) | 0.910/0.865 | 0.746/0.685 | 0.756/0.839 | 0.927/0.895 | 0.742/0.690 | 0.792/0.851 |

to false negatives (FN), ranging from 4:1 to 20:1 [2]. This suggests that the high rate of FP errors is the primary factor limiting the overall performance of the JudgeRail models.

In Figure 2, we illustrate the distribution of FP errors across different datasets and categories, with "P4" and "P5" denoting "Insult" and "Profanity", and "S6" and "S14" representing "Specialized Advice" and "Code Interpreter Abuse", respectively. The FP errors observed in ToxicChat are less severe than those found in toxic speech datasets. We can identify two key factors contributing to these FP errors: the taxonomic structure of the adopted label system and the quality of the provided labels. Upon analyzing the FP errors from HateCheck and HateXplain, a considerable number of samples, as exemplified in Table 1 in the appendix, were categorized by Gemma2(JR) under "Profanity" according to the Perspective API label system, despite being labeled as non-toxic or not harmful. Therefore, we attribute the discrepancies between the JudgeRail models' detection outcomes and the human-annotated "ground-truth" labels to the relatively high degree of semantic ambiguity inherent in the category designations of "Insult" and "Profanity".

We then adopted the label system of OpenAI Moderation API, which has less ambiguous semantic implications, as shown in Table 1. Table 3 show that, switching categories led to a decrease in F1 scores. However, the accuracy scores, which measure the ratio of correctly classified samples, slightly improved for JudgeRail models on the HateXplain and OpenAI moderation datasets. These changes can be attributed to a reduction in FP errors but an increase in FN errors when switching from the Perspective API's harmful categories to those of the OpenAI Moderation API. Figure 3 illustrates the change in the ratios between FP and FN errors before and after the label system switch for three JudgeRail LLMs. As expected, we observe an exchange between FP and FN errors for

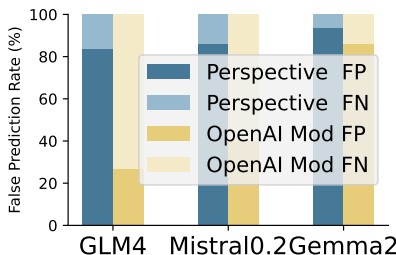

Figure 3: Change in the ratios between FP and FN errors on HateCheck before and after switching label systems.

both GLM4(JR) and Mistral0.2(JR), while FP errors remain a prominent issue for Gemma2(JR). These results indicate that changing the label system indeed influences detection performance, as expected. In JudgeRail, this can be adjusted by simply editing the text segment reserved for the label system. However, both label systems from the commercial moderation tools still exhibit a degree of ambiguity in their harmful taxonomies.

**The Impact of Label Quality**. We further focus on investigating the quality of the ground-truth labels provided for the identified FP samples. Specifically, we employed GPT-4 to re-label the FP samples collected from Gemma2(JR) on HateCheck. We asked GPT-4 to determine whether a sample labeled by Gemma2(JR) as "Insult" or "Profanity" was actually harmful, and used its assessment as the new ground-truth. As shown in Figure 4, the F1 scores of most evaluated models increase with the GPT-4 re-labeled samples. In particular, all JudgeRail LLMs exhibit more significant performance improvements, with Gemma2(JR) achieving the best performance on the HateCheck dataset. These results highlight the importance of label quality. Additionally, the OpenAI moderation dataset includes samples with fine-grained labels that align with the taxonomy of OpenAI Moderation API. This allowed the comparison of the fine-grained accuracy between Gemma2(JR) and OpenAI Moderation API. Specifically, we evaluated each sample from the OpenAI moderation dataset to determine if its ground-truth label matched the results of both Gemma2(JR) and OpenAI Moderation API, thereby obtaining the overall accuracy. The results, presented as the light blue bars in Figure 5, show that Gemma2(JR) achieved better detection accuracy.

---

[2]The specific FP and FN counts are detailed in the appendix.

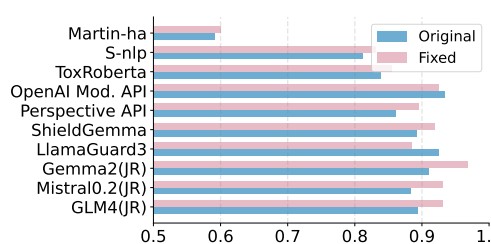

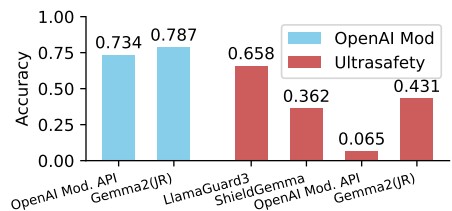

Figure 4: F1 scores before and after re-labeling with GPT-4 on HateCheck. "Original" denotes the original F1 scores, while "Fixed" denotes the F1 scores calculated with GPT-4's labels.

Figure 5: A comparison of detection accuracy between Gemma2 (JR) and the OpenAI Moderation API (left), and in detecting jailbreak prompts from Ultrasafety among Gemma2 (JR) and other models (right).

Table 4: Few-shot calibration performance (FP/FN/F1) for Gemma2(JR). "Base" denotes no calibration. "Individual" and "All" denote sampling FPs from individual and all datasets, respectively.

| Dataset | Base | | | Individual | | | All | | |
|---|---|---|---|---|---|---|---|---|---|
| | FP | FN | F1 | FP | FN | F1 | FP | FN | F1 |
| HateCheck | 472 | 31 | 0.910 | 414 | 47 | 0.916 | 472 | 35 | 0.909 |
| OpenAI Mod | 167 | 103 | 0.756 | 170 | 83 | 0.776 | 185 | 77 | 0.773 |

**Few-shot calibration**. To further improve the detection performance of JudgeRail models, we incorporate few-shot calibration examples into the prompting framework, as described in Section 3.1. Given that FP errors dominate the errors of Gemma2(JR), our primary objective is to introduce few-shot FP examples to calibrate its detection results. We constructed a pool of FP samples by collecting such errors from all datasets. We then conducted multiple sampling iterations, selecting 2, 4, or 8 samples from this pool to include in the JudgeRail prompting framework as calibration examples. However, since the HateXplain dataset, with 20K samples, contributes the majority of FP samples to the pool, random sampling fewer than 10 samples would essentially rely on HateXplain as the primary source for calibration. Therefore, we also sampled FP examples from each dataset individually and reported the best performance. Table 4 presents the experimental results obtained from the HateCheck and OpenAI moderation datasets [3]. We observe that on most datasets, sampling from individual datasets ("Individual") is more effective in reducing either FP or FN errors compared to sampling from the entire FP pool ("All").

### 4.4 PERFORMANCE FOR DETECTING ADVANCED JAILBREAK PROMPTS

We further evaluate Gemma2(JR) for its performance in detecting advanced jailbreak prompts, such as those provided by the UltraSafety dataset (Guo et al., 2024), which includes long jailbreak prompts for role-playing scenarios. The evaluation results for LlamaGuard3, ShieldGemma, Gemma2(JR), and OpenAI Moderation API are presented in Figure 5. Our observations indicate that LlamaGuard3 outperformed the other models by a large margin, while Gemma2(JR) achieved the second-best performance, surpassing ShieldGemma. Given that both Gemma2(JR) and ShieldGemma share the same foundation LLM, these results suggest that JudgeRail is effective in stimulating general-purpose LLMs to defend against more sophisticated jailbreak prompts.

### 4.5 IMPACT OF LOGIT RECTIFICATION AND MODEL PRECISION ON DETECTION LATENCY

In Figure 6, we illustrate the acceleration effect on detection achieved by adopting logit rectification. Compared to using the "soft" prompt defined in JudgeRail for output regulation, logit rectification significantly reduces processing latency for all three JudgeRail LLMs by ensuring a valid output format. The results in Table 2 also show that JudgeRail has a significant advantage over simple prompting in terms of latency. To further assess the practicality of JudgeRail, we evaluated the performance of 4-bit quantized LLMs under our framework, as well as the 4-bit versions of Llam-

---

[3]The results from other datasets are provided in the appendix.

Table 5: Performance of LLM-based detection models evaluated with BF16 and INT4 precision. We report accuracy scores for AdvBench and F1 scores for other datasets. Latency is measured as the average time in seconds required to process one sample.

| Model | Dataset | | | | | Latency | Mem. |
|-------|---------|---|---|---|---|---------|------|
| | HateCheck | HateXplain | OpenAI Mod | Advbench | ToxicChat | | |
| LlamaGuard3-BF16 | **0.926** | 0.720 | 0.791 | 0.979 | 0.497 | 0.159 | 26G |
| LlamaGuard3-INT4 | 0.885 | 0.689 | 0.780 | 0.983 | 0.489 | 0.234 | 9G |
| ShieldGemma-BF16 | 0.892 | 0.729 | **0.794** | 0.612 | 0.684 | 0.191 | 31G |
| ShieldGemma-INT4 | 0.892 | 0.728 | 0.790 | 0.477 | 0.675 | 0.260 | 12G |
| Gemma2(JR)-BF16 | 0.916 | 0.746 | 0.776 | **0.996** | 0.618 | 0.094 | 21G |
| Gemma2(JR)-INT4 | 0.920 | **0.755** | 0.748 | 0.992 | **0.687** | 0.154 | 11G |

aGuard3 and ShieldGemma[4]. We monitored the running memory (as shown in the "Mem." column of Table 5) to validate the success of the configuration.

As shown in Table 5, most LLMs with 4-bit precision maintain rather stable performance across most datasets. However, ShieldGemma experiences a noticeable performance degradation on the AdvBench dataset when transitioning from BF16 to INT4 precision. In contrast, on the ToxicChat dataset, Gemma2(JR) with INT4 precision outperforms its BF16 counterpart and achieves the best performance. These results indicate that open-source LLMs equipped with JudgeRail can be effectively deployed at lower precision while maintaining

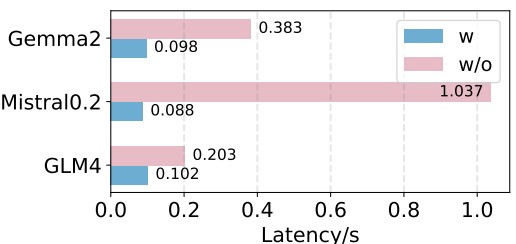

Figure 6: Compare latency with (label w) or without (label w/o) logit rectification

their detection performance. We also observed that all LLM-based models experienced nearly twice the latency compared to their BF16 versions. This increase in latency may be attributed to the current limitations in computational efficiency optimizations for 4-bit quantized models. As lower-bit LLMs are anticipated to be better optimized, the practicality of JudgeRail can be further boosted.

## 5 LIMITATION

To ensure fast detection, we use a relatively simple in-context learning mechanism, which may limit performance improvement. In future work, we will explore more complex mechanisms, such as Retrieval-Augmented Generation (RAG). Moreover, we have shown that existing label systems have some degree of semantic ambiguity, which limits detection performance. This motivates our future work on designing a more refined label system with better-separated semantic representations. Additionally, note that the detection capabilities of JudgeRail essentially depend on its underlying LLM. This indicates that one needs to perform model selection instead of using arbitrary LLMs. However, JudgeRail also benefits from this as the underlying LLM evolves in its capability.

## 6 CONCLUSION

This paper introduces the JudgeRail framework, which effectively and efficiently adapts open-source LLMs into harmful text detectors. We have thoroughly investigated the influence of the label system, in-context few-shot calibration examples, and a novel logit rectification method. We have evaluated three open-source LLMs equipped with JudgeRail, as well as several LLM-based and conventional moderation tools, on five datasets that encompass toxic speech data and jailbreak prompts. Our experiments demonstrate that LLM-based models achieve significantly better performance than conventional detectors, while LLMs with JudgeRail are competitive with fine-tuned moderation LLMs. The results also show that LLMs with JudgeRail require approximately half the time needed by LlamaGuard3 and ShieldGemma to process a sample on average.

---

[4]We configured all models to their 4-bit versions by setting "load_in_4bit=True".

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
