# OpenReview forum: "JudgeRail: Harnessing Open-Source LLMs for Fast Harmful Text Detection with Judicial Prompting and Logit Rectification"
_ICLR.cc/2025/Conference — Submitted to ICLR 2025_

### Official Review · Reviewer_p6GN · 2024-11-03

**Soundness:** 3
**Presentation:** 2
**Contribution:** 3
**Rating:** 5
**Confidence:** 5

**Summary:**

The paper introduces JudgeRail, a framework designed to enhance harmful text detection using open-source large language models (LLMs) without requiring extensive fine-tuning. By leveraging "judicial prompting" and a novel "logit rectification" technique, JudgeRail ensures accurate text classification and significantly reduces detection latency. Evaluated against established tools like OpenAI’s Moderation API and specialized models like LlamaGuard3 and ShieldGemma, JudgeRail-equipped LLMs demonstrated competitive performance while achieving faster processing times (only 46-55% of the time required by other advanced models).

**Strengths:**

1) Introducing label systems, novel logit rectification method and calibration is really interesting and found to be helpful.

2) The evaluation and baselines includes the state of the art models.

3) The latency seems be a strength of the proposed model.

**Weaknesses:**

1) No limitations section

2) No comparison with prompt detection techniques. Prompt detectors are simple and easy to integrate as well as latency is quite low.

3) Lack of enough baselines. Readers and researchers will be interested in comparison with prompting techniques which are much simpler to execute like baselines with only chain of thought prompting and other advanced prompting techniques.

4) The datsets implemented looks irrelevant hateful prompts differ from hate speech and hatexplain. please refer[1],[2] and it would be great if you can implement those.

5) The novel method cannot be implemented on the GPT models which is the model used by most. This makes the model cannot be utilised by many who are dependent on GPT models.

[1] Realtoxicityprompts: Evaluating neural toxic degeneration in language models
[2] Efficient Detection of Toxic Prompts in Large Language Models

**Questions:**

1) Can you implement the proposed model on the GPT models.

2) Can this be applicable for vison models as well like text to image generation. please write a section in appendix.

3) Is the proposed method much faster than prompt detection techniques?

4) Why there is no evaluation comparison with prompting techniques? Though they might look simple readers wouldlike to their evaluation metrics as well.

---

> ### Author Response · Authors · 2024-11-20
> **Response to Reviewer p6GN (Part 1)**
>
> Thank you for your valuable feedback.
>
> # Weakness 1
>
> Thank you for your suggestion. We will discuss the limitations from the following points: First, due to latency considerations, our current in-context learning mechanism is relatively simple. More complex mechanisms, such as Retrieval-Augmented Generation (RAG), will be explored in future work. Additionally, we have shown that the label system has a clear impact on detection performance, while existing label systems have some degree of semantic ambiguity. This inspires us to consider designing a more refined label system with better-separated semantic representations. Finally, while the detection capabilities of JudgeRail essentially depend on its underlying LLM, JudgeRail also benefits from this as the underlying model evolves in its capability.
>
> # Weakness 2&4
>
> Thank you for the reference, and we will include it in our related work. Following your suggestion, we employed the RealToxicityPrompts [1] dataset for evaluating several models used in our paper and the recommended ToxicDetector [2]. Since the ToxicChat dataset is also a toxic prompt dataset, we have included the test results for both datasets, as shown in the following table (reporting F1-score with the best performance):
>
> | Dataset\method      | Gemma2(JR) | LlamaGuard3 | ShieldGemma | Perspective API(Reproduce\Report) | ToxicDetector[2] |
> | ------------------- | ---------- | ----------- | ----------- | --------------------------------- | ---------------- |
> | RealToxicityPrompts | 0.617      | 0.231       | 0.482       | 0.685\0.8674                      | 0.9628           |
> | ToxicChat           | 0.687      | 0.497       | 0.684       | 0.250                             | -                |
>
> As shown in the above table, the reported performance of ToxicDetector outperforms other models on the RealToxicityPrompts by a large margin. We noticed from [2] that ToxicDetector is trained and evaluated on this RealToxicityPrompts dataset. As we have done for SplineLLM to respond to the second reviewer's comment, we would like to further evaluate the generalization performance of ToxicDetector. However, we find that this method has not been released to the public, making it difficult to evaluate ToxicDetector [2] on other datasets.
>
> In the meantime, we have noticed that both [2] and our work have adopted Perspective API for comparison, and the performance of Perspective API on RealToxicityPrompts has been reported. We have evaluated Perspective API on RealToxicityPrompts with the same sample size to reproduce the reported result. However, as shown in the above table, our reproduced performance is significantly worse than the reported result. Such inconsistency makes it difficult to draw a reasonable comparison with the recommended ToxicDetector.
>
> Nevertheless, we have evaluated our proposed method, along with other two LLM-based moderation models, on this RealToxicityPrompts dataset. We have also presented our previous evaluation results obtained on ToxicChat in the above table, as both datasets contain prompt-type harmful text samples.

---

> ### Author Response · Authors · 2024-11-20
> **Response to Reviewer p6GN (Part 2)**
>
> # Weakness 3
>
> As we have also responded to the 3rd reviewer's comments, we employed two simple prompt methods from several related studies [3] [4] to conduct experimental comparisons. Given that these methods are aimed at hate speech and toxic content, and due to time constraints, we selected three harmful text datasets for evaluation, as shown below (reporting F1-score):
>
>
>
> | Gemma2        | HateCheck | HateXplain | OpenAI Mod | Latency(s) |
> | ------------- | --------- | ---------- | ---------- | ---------- |
> | simple_COT[3] | 0.905     | 0.711      | 0.693      | 7.310      |
> | simple[4]     | 0.887     | 0.712      | 0.730      | 7.392      |
> | JudgeRail     | 0.910     | 0.746      | 0.756      | 0.098      |
>
> This new evaluation result shows that simple prompting can relatively shape an LLM to obtain satisfactory performance. Meanwhile, JudgeRail maintains superior performance across all datasets and has a significant advantage in terms of latency.
>
>
>
> # Weakness 5
>
> Actually, our proposed JudgeRail can be applied to GPT models. We chose not to include GPT models in our original experiments since we aim to propose a solution that uses open-source LLMs, which have comparable model sizes to existing LLM-based moderation tools, such as LlamaGuard3 and ShieldGemma. Nevertheless, to better present the generalization capability of JudgeRail, we have evaluated the performance of using GPT4 with JudgeRail on three datasets and present the results in the following table. We selected these datasets since their data sizes are relatively small and cause relatively lower costs of calling the GPT4 API.
>
> | Dataset    | AdvBench | HateCheck | OpenAI Mod |
> | ---------- | -------- | --------- | ---------- |
> | GPT4(JR)   | 0.988    | 0.821     | 0.732      |
> | Gemma2(JR) | 0.992    | 0.910     | 0.756      |
>
> As shown in the above table, GPT4 equipped with JudgeRail -- GPT4(JR) obtains comparable performance to our best-performing Gemma2(JR) on AdvBench and OpenAI Moderation datasets, while performing worse on the HateCheck dataset. By examining the samples mistakenly classified by GPT4(JR), we find that, while HateCheck primarily focuses on hate speech, some of its samples labeled as Non-hate still contain offensive materials. This type of content is often recognized as harmful by GPT4. This aligns with our previous findings mentioned in the paper regarding the ambiguity in the label system and the inaccuracies in dataset labels.

---

> ### Author Response · Authors · 2024-11-20
> **Response to Reviewer p6GN (Part 3)**
>
> # Question 1
>
> JudgeRail can be implemented on the GPT models and  we have presented the evaluation results for GPT4 equipped with JudgeRail in the response above.
>
>
> # Question 2
>
> Thank you for your valuable question. We would like to demonstrate that JudgeRail can be applied to text-to-image models. We have evaluated JudgeRail with several text-to-image prompt datasets from prior research [5]. The results are shown below (reporting accuracy; the Template prompts dataset contains 30 samples, the Lexica prompts dataset contains 404 samples, and the other two datasets each contain 500 samples.):
>
> | Dataset   | Template prompts | Lexica prompts | MS COCO prompts | 4chan prompts |
> | --------- | ---------------- | -------------- | --------------- | ------------- |
> | Gemma(JR) | 1.00             | 0.38           | 0.99            | 0.94          |
>
> We will provide a description on utilizing JudgeRail for recognizing harmful prompts used for attacking text-to-image generation models and present the updated evaluation results in the appendix.
>
>
> # Question 3
>
> Since we cannot reproduce ToxicDetector as previously mentioned, we can only compare to the latency reported in [2].
> Specifically, we first referenced the results from [2] for the Perspective API, which reports a latency of 0.8 seconds. This latency is close to our measurements for the latency of calling Perspective API, which is around 1 second. Based on this consistent latency result, we observed that the ToxicDetector reported in [2] has a latency of 0.078 seconds, and our average latency ranges from 0.088 to 0.102 seconds. This demonstrates that ToxicDetector is faster than the proposed method by around 13%.
>
> # Question 4
>
> Please refer to the response for weakness 3.
>
>
>
> [1] Realtoxicityprompts: Evaluating neural toxic degeneration in language models
>
> [2] Efficient Detection of Toxic Prompts in Large Language Models
>
> [3] Yongjin Yang, Joonkee Kim, Yujin Kim, Namgyu Ho, James Thorne, and Se-Young Yun. 2023. [HARE: Explainable Hate Speech Detection with Step-by-Step Reasoning](https://aclanthology.org/2023.findings-emnlp.365). In *Findings of the Association for Computational Linguistics: EMNLP 2023*, pages 5490–5505, Singapore. Association for Computational Linguistics.
>
> [4] He X, Zannettou S, Shen Y, et al. You only prompt once: On the capabilities of prompt learning on large language models to tackle toxic content[C]//2024 IEEE Symposium on Security and Privacy (SP). IEEE, 2024: 770-787.https://arxiv.org/pdf/2308.05596
>
> [5]Yiting Qu, Xinyue Shen, Xinlei He, Michael Backes, Savvas Zannettou, and Yang Zhang. 2023. Unsafe Diffusion: On the Generation of Unsafe Images and Hateful Memes From Text-To-Image Models. In Proceedings of the 2023 ACM SIGSAC Conference on Computer and Communications Security (CCS '23). Association for Computing Machinery, New York, NY, USA, 3403–3417. https://doi.org/10.1145/3576915.3616679

---

### Official Review · Reviewer_yCFh · 2024-11-04

**Soundness:** 3
**Presentation:** 3
**Contribution:** 3
**Rating:** 8
**Confidence:** 3

**Summary:**

This paper presents JudgeRail, a framework designed to adapt LLMs for detecting harmful content. The authors conduct extensive experiments comparing the performance of open-source LLMs with JudgeRail, against traditional harmful content detection models and commercial APIs. They introduce a logit rectification method to refine LLM outputs, ensuring more valid classifications and reducing latency. Results show that open-source LLMs equipped with JudgeRail perform comparably to commercial APIs and outperform conventional detection methods.

**Strengths:**

- The research question is well-motivated, emphasizing the importance of detecting harmful content and preventing prompt jailbreaking in publicly deployed models. This paper effectively explores how to adapt open-source LLMs for better harmful content detection. In a landscape dominated by commercial APIs, it is crucial to investigate scalable and effective methods for open-source models. The results show that while open-source LLMs may not surpass commercial solutions in all respects, they can perform comparably, highlighting their value in content moderation efforts.

- They conduct extensive experiments to provide a robust comparison between open-source LLMs and both traditional and commercial detection models. Also, they explore the validity of their design choices very meticulously.

**Weaknesses:**

- The novel logit rectification method has shown effectiveness on a limited set of examples. However, it is difficult to assess its overall impact on the framework's performance. The paper is missingcomparisons using simple prompts on the LLMs and ablation studies that evaluate performance with and without the logit rectification method, as these analyses could provide clearer insights into its contribution.

- While the paper is generally easy to understand, the experiments section is densely written, making it challenging to follow all the observations. For example, Section 4.3 would benefit from the inclusion of small headings or bolded paragraphs headings to better organize and group the observations, which would significantly enhance readability.

Minor comment:
- You use "a LLM" in many places (lines 183, 192 etc.), but I think it should be "an LLM". Please check with a native speaker and make correction if required

**Questions:**

- Do you have any insights on how these open-source models operate without the JudgeRail framework? Are the LLMs capable enough with simple prompts, and does JudgeRail generally enhance their performance?

---

> ### Author Response · Authors · 2024-11-20
> **Response to Reviewer yCFh**
>
> Thank you for your valuable feedback and suggestions.
>
> # Weakness 1
>
> Thank you very much for your valuable suggestions. As we have also responded to the 2nd reviewer's comments, for the comparative experiment with and without the logit rectification, we will continue to test with larger scales, spanning 500 to 1000 samples, with an equal number of samples randomly sampled from all five datasets used in this work. However, we would like to highlight that we introduce the logit rectification mechanism primarily to extract valid formatted decisions, even when an LLM generates outputs that do not conform to JudgeRail's output format specification. In other words, the logit rectification mechanism serves as an efficient and effective error-handling mechanism. Therefore, we used the 100 samples to validate that using logit rectification, compared to recognizing model decisions from its unstructured generated content via sophisticated parsing mechanisms, has minimal impact on the moderation performance.
>
> We followed your suggestion and employed two simple prompt methods from several related studies [1] [2] to conduct experimental comparisons. Given that these methods are aimed at hate speech and toxic content, and due to time constraints, we selected three harmful text datasets for evaluation, as shown below (reporting F1-score):
>
> | Gemma2        | HateCheck | HateXplain | OpenAI Mod | Latency(s) |
> | ------------- | --------- | ---------- | ---------- | ---------- |
> | simple_COT[1] | 0.905     | 0.711      | 0.693      | 7.310      |
> | simple[2]     | 0.887     | 0.712      | 0.730      | 7.392      |
> | JudgeRail     | 0.910     | 0.746      | 0.756      | 0.098      |
>
> This new evaluation result shows that simple prompting can relatively shape an LLM to obtain satisfactory performance. Meanwhile, JudgeRail maintains superior performance across all datasets and has a significant advantage in terms of latency.
>
>
>
> [1] Yongjin Yang, Joonkee Kim, Yujin Kim, Namgyu Ho, James Thorne, and Se-Young Yun. 2023. [HARE: Explainable Hate Speech Detection with Step-by-Step Reasoning](https://aclanthology.org/2023.findings-emnlp.365). In *Findings of the Association for Computational Linguistics: EMNLP 2023*, pages 5490–5505, Singapore. Association for Computational Linguistics.
>
> [2] He X, Zannettou S, Shen Y, et al. You only prompt once: On the capabilities of prompt learning on large language models to tackle toxic content[C]//2024 IEEE Symposium on Security and Privacy (SP). IEEE, 2024: 770-787.https://arxiv.org/pdf/2308.05596
>
> # Weakness 2 and writing
>
> Thank you for your valuable feedback regarding the organization of the experiments section. We will revise this section by adding clear subheadings or bolded paragraphs headings to better structure the observations and improve clarity. The typo "a LLM " will also be fixed to "an LLM". Thanks again for your kind advice.
>
>
>
> # Questions:
>
> Thank you for raising this question, and we are glad to provide further clarification on our thoughts.
>
> Since an open-source LLM trained and released for generic text generation tasks is commonly pre-trained on a vast amount of data, we believe the model already understands certain common knowledge, such as those used in composing harmful text. Indeed, by examining most harmful text datasets, we find that the content moderation task primarily involves common sense and everyday language, with rather limited requirements for domain-specific knowledge. As such, we think these open-source models may be able to obtain satisfactory performance even without the JudgeRail framework.
>
> According to the result shown in the above table, as expected, an LLM equipped with simple prompting techniques demonstrates good moderation performance, while JudgeRail demonstrates a notable improvement on the dataset and also has a significant advantage in terms of latency. However, during this newly added experiment, we found that an LLM often does not output according to the format instructed in the prompt. This highlights the need for a mechanism, such as the proposed logit rectification, to complement the prompts, thereby better converting an LLM into a powerful content moderation tool.

---

> > ### Comment · Reviewer_yCFh · 2024-11-25
> >
> > Thanks to the authors for their detailed responses and clarifications. Considering the responses provided by authors to all reviews, I continue to believe that this paper has merits. Therefore, I will maintain my current score."

---

> ### Author Response · Authors · 2024-11-25
> **Thank you for your feedback**
>
> We appreciate your comments and will revise our manuscript accordingly.

---

### Official Review · Reviewer_4vQg · 2024-11-05

**Soundness:** 3
**Presentation:** 2
**Contribution:** 2
**Rating:** 5
**Confidence:** 4

**Summary:**

The paper presents a moderation framework for harmful text detection using open-source LLMs. First, the framework  uses prompting based on jurisdictal principles (assigning a judge role, using chain-of-thought prompting) and secondly, restrictis the logit output to a predetermined few-shot learned set of labels that are based on PerpepeciveAI, the OpenAI Moderation API and LlamaGuard3. The authors claim this is more efficient while performing almost on par with the closed models’ APIs.

**Strengths:**

The paper introduces a simple and low-cost approach using prompting as a content moderation technique, along with controlled decoding. By adapting the prompting strategy of open LLMs models to the schema of the baseline APIs also including LlamaGuard3 and ShieldGemma, along with controlled decoding the authors achieve competitive performance without extra fine-tuning. This is interesting in spite of its simplicity. They provide a multi-faceted evaluation of false-positive classification.

The paper is generally well-written.

**Weaknesses:**

While multiple aspect of the false-positive ratio are evaluated, I have several concerns regarding the evaluation.

1. The reated work section mentions two recent approaches, SplineLLM and RigorLLM, which JudgeLLM is not compared to (only what the authors call conventional models based on BERT (martin-ha, toxberta, s-nlp, all bert-based); I find this comparison important. How does this approach differ from those two?

2. Label set and constrained decoding:
- Lines 305-310: E.,g, “For Perspective API, we converted its multi-label detection results into binary classification ”: how does it perform using multi-label output?
- How much does the approach depend on the decoding vocabulary?
    - I find a more detailed evaluation of the logit distribution important:
	 - different decoding vocabulary sizes
	 - a comparison to the performance using the overall logit distribution, without restrictions
- The authors used only 100 samples to create the decoding vocabulary (logit distribution): how much does performance depend on the sample size and does it change with more/less samples?

Writing:
The phrasing in some parts is unneccessarily strong, e.g.: Lines 20-22: "accurately interprets an LLM’s classification intent, rigorously controls its output format, and significantly accelerates detection"

**Questions:**

A more general question:
Interestingly, this simple approach performs on par with closed models' APIs and also fine-tuned ones such as  LlamaGuard3, while also being much more cost-efficient. What is your intuition on that? E.g, how much is it depending much on the task/datsets?

---

> ### Author Response · Authors · 2024-11-20
> **Response to Reviewer 4vQg (Part 1)**
>
> Thank you for your review and the insightful comments.
>
> # Weakness 1
>
> We initially focused on comparing with content moderation methods that are more suitable for practical use cases, leading us to select moderation tools and models (such as Perspective API, OpenAI Moderation API, ShieldGemma, and LlamaGuard3) released by companies with real moderation demands. We acknowledge the importance of empirically comparing with research-oriented moderation methods. Following your suggestion, we have attempted to include comparisons with SplineLLM and RigorLLM.
>
>
> SplineLLM proposes to utilize the internal representations of LLMs to characterize a given prompt and generation. In SplineLLM, content moderation is a subtask, where an instance of SplineLLM is trained and tested on the same dataset, Jigsaw [1]. Meanwhile, RigorLLM integrates several different processes and components into its proposed framework, including energy-based hidden embedding data augmentation, optimization-based prompt suffix generation, and a fusion-based model combining robust KNN with LLMs. This indicates that RigorLLM has a relatively more sophisticated implementation. Indeed, when evaluating RigorLLM, we encountered difficulties in deploying RigorLLM, facing running and configuration issues. Due to limited time, we have not yet obtained the results. We will continue to work on reproducing it.
>
>
>
> In contrast, we have successfully reproduced the SplineLLM approach and included the results in our experiments, as shown in the following table in markdown format. For AdvBench, we report the accuracy, while for other datasets, we report the F1-score. The latency is measured in seconds.
>
>
> | Model/Dataset   | HateCheck | HateXplain | OpenAI Mod | AdvBench | ToxicChat | Latency |
> | --------------- | --------- | ---------- | ---------- | -------- | --------- | ------- |
> | Martin-ha       | 0.592     | 0.511      | 0.504      | 0.000    | 0.114     | 0.001   |
> | ToxRoberta      | 0.839     | 0.685      | 0.612      | 0.210    | 0.274     | 0.002   |
> | S-nlp           | 0.812     | 0.664      | 0.684      | 0.019    | 0.265     | 0.001   |
> | Perspective API | 0.862     | 0.683      | 0.701      | 0.054    | 0.250     | 1.000   |
> | OpenAI Mod. API | 0.934     | 0.744      | 0.790      | 0.104    | 0.254     | 1.030   |
> | LlamaGuard3     | 0.926     | 0.720      | 0.791      | 0.979    | 0.497     | 0.159   |
> | ShieldGemma     | 0.892     | 0.729      | 0.794      | 0.612    | 0.684     | 0.191   |
> | SplineLLM       | 0.815     | 0.667      | 0.481      | 0.892    | 0.139     | 0.063   |
> | GLM4(JR)        | 0.894     | 0.719      | 0.714      | 0.729    | 0.385     | 0.102   |
> | Mistral0.2(JR)  | 0.884     | 0.706      | 0.676      | 0.950    | 0.586     | 0.088   |
> | Gemma2(JR)      | 0.910     | 0.746      | 0.756      | 0.992    | 0.584     | 0.098   |
>
>
> As previously mentioned, SplineLLM is trained and evaluated on the same dataset.
> As shown in the above table, the generalization of SplineLLM is limited when evaluated across different datasets. Furthermore, the performance of SplineLLM is more closely aligned with Martin-ha, ToxRoberta, and S-nlp, with a significant difference being its performance on AdvBench, which approaches that of several LLM-based moderation solutions. Due to the simplicity of SplineLLM, its serving latency is noticeably lower than that of LLM-based moderation solutions, yet still significantly higher than Bert-based moderation solutions.

---

> ### Author Response · Authors · 2024-11-20
> **Response to Reviewer 4vQg (Part 2)**
>
> # Weakness 2
>
> (1) We chose to evaluate Perspective API by its binary classification performance, as Perspective API does not have a released, associated labeled multi-label dataset for fine-grained comparison. Therefore, to evaluate and compare JudgeRail in a multi-label setting, we selected the OpenAI Moderation API, which offers a labeled multi-label dataset.
>
>
>
> (2) We apologize for the misleading presentation. We would like to specify that we introduce the logit rectification mechanism primarily to extract valid formatted decisions, even when an LLM generates outputs that do not conform to JudgeRail's output format specification. In other words, the logit rectification mechanism serves as an error-handling mechanism, avoiding the need for post-output parsing-based error handling methods. Therefore, we used the 100 samples to validate that using logit rectification, compared to recognizing model decisions from its unstructured generated content via sophisticated parsing mechanisms, has minimal impact on the moderation performance.
>
>
> (3) Following the above clarification, we would like to specify that we did not use the selected 100 samples to build a decoding vocabulary. The decoding vocabulary size depends on the label system used. For example, when using the Perspective API label system, the vocabulary can range from "0"-"6". When using the OpenAI Moderation API label system, the vocabulary can range from "0"-"9" and "A"-"E".
>
> For the comparative experiment with and without the logit rectification, we will continue to test with larger scales, spanning 500 to 1000 samples, with an equal number of samples randomly sampled from all five datasets used in this work.
>
> We will clarify the aforementioned misleading points in Section 3.2 of the revised manuscript.
>
>
>
> # Writing:
>
> Thank you for the advice. We will revise the description in the corresponding section of the manuscript.
>
> # Question:
>
> Thank you for raising this question, and we are glad to provide further clarification on our thoughts.
>
> First, we believe that an LLM trained and released for generic text generation tasks is commonly pre-trained on a vast amount of data, enabling the model to already understand certain common knowledge, such as those used in composing harmful text. Indeed, by examining most harmful text datasets, we find that the content moderation task primarily involves common sense and everyday language, with rather limited requirements for domain-specific knowledge.
>
> Furthermore, by surveying other literature, we have noticed that some prior research[2] has highlighted the questionable effectiveness of fine-tuning LLMs, as it primarily changes the style of the LLM's output rather than its knowledge.
>
> The combination of these latter two observations and the former intuition guided our design of the prompting framework. Our goal is to leverage this common sense more effectively. Therefore, we chose to assign an LLM a commonly known Judge character, with the commonly acquired principle of "presumption of innocence", and validated our design through thorough experiments.
>
>
>
> [1] Adams, C., Jeffrey, S., Julia, E., Lucas, D., Mark, M.,Nithum, and Will, C. Toxic comment classification challenge, 2017.
>
> [2]Lin B Y, Ravichander A, Lu X, et al. The unlocking spell on base llms: Rethinking alignment via in-context learning[C]//The Twelfth International Conference on Learning Representations. 2023.

---

> > ### Comment · Reviewer_4vQg · 2024-11-25
> >
> > Thank you very much for the response. After reading the comparison with different baselines, including existing prompt-based approaches as following by reviewer p6GN’s concerns (not mentioned in the original manuscript), I find many similar methods with comparative performance exist.
> > As I find that hampering the novelty of this approach mainly to low-latency (which could be mitigated using inference-focused tools such as vllm), I will keep my original score.

---

> > > ### Author Response · Authors · 2024-11-25
> > > **Thank you for your feedback**
> > >
> > > We appreciate your consideration of the comparisons with existing prompt-based approaches and the concerns raised by Reviewer p6GN. We would like to emphasize several key points:
> > >
> > > 1. While prompt engineering is a common practice, our work demonstrates that simple prompting alone has limited effectiveness in enhancing text moderation, as evidenced by the results presented in our response to Reviewer p6GN. This finding underscores the necessity for more sophisticated designs. In particular, we have conducted extensive experiments with the label system, which not only highlight its impact on moderation performance, an area rarely explored in existing literature, but also demonstrate the flexibility and efficiency (in terms of fine-tuning cost) for adapting our prompt framework to accommodate new moderation categories or requirements.
> > >
> > > 2. Our approach stands out due to its ability to achieve high performance with a single, streamlined prompt structure, which is closely integrated with our logit rectification mechanism. It enables a single round of detection with desirable performance. In contrast, other methods often require multiple LLMs or iterative rounds of reflection to achieve comparable performance, leading to considerable delays in processing individual text samples. Efficiency is crucial in real-world applications where latency and computational resources are critical factors.
> > >
> > > 3. Regarding low-latency, our logit rectification mechanism can operate in tandem with tools such as vLLM, rather than being exclusive. Moreover, we believe it is a generic method that can be applied to other classification-oriented tasks, achieving both output format control and acceleration simultaneously. Therefore, we maintain that this aspect represents a core advantage of our method.
> > >
> > > We believe these aspects collectively highlight the novelty and practical value of our approach. We appreciate your understanding and hope these clarifications provide additional context for your evaluation.

---

### Official Review · Reviewer_MwTd · 2024-11-07

**Soundness:** 2
**Presentation:** 2
**Contribution:** 2
**Rating:** 5
**Confidence:** 3

**Summary:**

This paper introduces JudgeRail, a framework designed to guide open-source language models in detecting harmful text through judicial prompting and a novel logit rectification technique. The study compares JudgeRail’s effectiveness and efficiency against existing moderation tools, showing that JudgeRail enhances detection accuracy and latency without requiring fine-tuning.

**Strengths:**

1. Significance: The paper tackles a critical issue, harmful text detection, which is increasingly urgent as LLMs are deployed widely in real-world applications.

2. Comprehensive Evaluation: The paper includes comparisons with state-of-the-art models and examines latency, a rarely explored aspect in text moderation studies, providing practical insights for real-world applications.

**Weaknesses:**

1. Insufficient Justification: The paper does not adequately justify the specific harmful categories used in the label system, which could limit the generalizability of its results. What are P1, P2, etc., and S1, S2, etc?

2. Lack of Novelty in Core Concept: Judge framework in content moderation is not new. There are some missed literature already explored this concept and implemented. Such as:

Mitchell L. Gordon, Michelle S. Lam, Joon Sung Park, Kayur Patel, Jeff Hancock, Tatsunori Hashimoto, and Michael S. Bernstein. 2022. Jury Learning: Integrating Dissenting Voices into Machine Learning Models. In Proceedings of the 2022 CHI Conference on Human Factors in Computing Systems (CHI '22). Association for Computing Machinery, New York, NY, USA, Article 115, 1–19. https://doi.org/10.1145/3491102.3502004

It would be important for the authors to justify their approach's novelty and difference with previous relevant work.

**Questions:**

It is not clear why the paper claims that "These findings suggest that text moderation tasks have lower requirements for high-precision computing compared to text generation tasks." Low precision is the results, how could it explain the requirement?
However, achieving acceptable results with lower precision does not inherently justify that text moderation has lower precision requirements.
Instead, it simply indicates that this particular framework, JudgeRail, was effective in the given tests. The paper would benefit from clarifying the distinction between observed outcomes and actual task requirements, and ideally, providing evidence or reasoning that explains why moderation tasks inherently need less precision compared to generation tasks.

---

> ### Author Response · Authors · 2024-11-20
> **Response to Reviewer MwTd**
>
> Thank you for your valuable comments.
>
> # Weakness 1
>
> We would like to clarify the design of our label system and will provide a more detailed description in the revised manuscript. To avoid arbitrary selection of harmful categories, we have adopted the categories defined by Perspective API  and LlamaGuard3, which offer more fine-grained classifications compared to other reviewed moderation tools.
>
> To be specific, the P1, P2, ..., P6 correspond to the Perspective API's harmful categories (in Table 1, where 1.Toxicity, 2.Severe toxicity, ..., 6.Threat). We incorporate all 6 categories in our label system when testing the content datasets.
> And S1, S2, ..., S14 correspond to LlamaGuard3's harmful categories (in Table 1, where 1.Violent crimes, 2.Non-violent crimes, ..., 14.Defamation).  We incorporate all 14 categories in our label system when testing the prompt datasets.
>
> # Weakness 2
>
> Thank you for providing this reference [1]. We acknowledge that in Jury Learning [1], dissenting voices are integrated by modeling individual annotators and allowing practitioners to define the jury composition. Additionally, we have identified another related work, Digital Juries [2], which proposes a civics-oriented approach for adjudicating content moderation cases. However, both approaches necessitate deploying multiple models, potentially consuming significant GPU memory resources and introducing a complex decision-making mechanism, which may result in additional latency.
>
> In contrast, we named our framework JudgeRail, inspired by the role-playing scheme commonly adopted in developing jailbreak prompts. While one can assign an LLM a harmful character, which may be challenging due to more mature safety alignment, we can also assign an LLM a helpful character to combat harmful content. This character must be fair, driving us to select the principle of "presumption of innocence," which naturally fits the "Judge" character and is a common-sense principle that most LLMs can understand and follow.
>
> Moreover, while sharing the spirit of introducing knowledge from the judicial system, practicality is a key consideration in JudgeRail. This motivates us to design a generic prompt framework that works with individual LLMs and incorporates the logit rectification mechanism, which accelerates processing and efficiently handles out-of-scope generation issues.
>
> We will revise the manuscript to more clearly articulate our novelty and incorporate the newly provided references.
>
> # Question
>
> Thank you for pointing out the less clarified statement. We will rephrase our presentation to highlight our observation regarding low-precision models. Specifically, our evaluation results, collected from three LLMs including Gemma2(JR), ShieldGemma, and LlamaGuard3, indicate that adopting their low-precision counterparts introduces rather limited performance impact. We find this observation intriguing, as it contrasts with studies [3,4] that have demonstrated the crucial role of model precision in generative tasks. This differing impact on model performance, driven by low-precision models, leads us to wonder that decision-making moderation tasks may have distinct requirements for model precision. We will provide a clearer presentation in the revised manuscript.
>
> [1] Mitchell L. Gordon, Michelle S. Lam, Joon Sung Park, Kayur Patel, Jeff Hancock, Tatsunori Hashimoto, and Michael S. Bernstein. 2022. Jury Learning: Integrating Dissenting Voices into Machine Learning Models. In Proceedings of the 2022 CHI Conference on Human Factors in Computing Systems (CHI '22). Association for Computing Machinery, New York, NY, USA, Article 115, 1–19. https://doi.org/10.1145/3491102.3502004
>
> [2] Jenny Fan and Amy X. Zhang. 2020. Digital Juries: A Civics-Oriented Approach to Platform Governance. In Proceedings of the 2020 CHI Conference on Human Factors in Computing Systems (CHI '20). Association for Computing Machinery, New York, NY, USA, 1–14. https://doi.org/10.1145/3313831.3376293
>
> [3]Li S, Ning X, Wang L, et al. Evaluating quantized large language models[J]. arXiv preprint arXiv:2402.18158, 2024.
>
> [4] Gong Z, Liu J, Wang J, et al. What Makes Quantization for Large Language Model Hard? An Empirical Study from the Lens of Perturbation[C]//Proceedings of the AAAI Conference on Artificial Intelligence. 2024, 38(16): 18082-18089.

---

> > ### Comment · Reviewer_MwTd · 2024-11-24
> > **Thanks the authors for their clarification**
> >
> > Thanks the authors for their clarification. After reviewing the authors' response and other reviewers' review, I would like to keep my original score.

---

### Author Response · Authors · 2024-11-23
**General Response (Part 1)**

We appreciate all reviewers for their valuable comments, suggestions, and questions. We have revised our manuscript accordingly to address these points. Additionally, a diff file is included in the supplementary material for the reviewers' convenience.

---

### Meta-Review · Area_Chair_Z5cT · 2024-12-18

**Metareview:**

**Summary:**

The authors propose a LLM-based content moderation framework (JudgeRail) that prompts LLMs to assume the persona of a judge in detecting harmful text. This approach removes the need for additional fine-tuning, which the authors show leads to an efficient and effective system compared against other well-known content moderation solutions like LlamaGuard and ShieldGemma.

**Strengths:**

- The paper offers a solution to an important, timely problem (automated content moderation detection)

- The discussion of detection latency and 4-bit comparison in the paper is interesting, and I believe the authors have a valid point about inference-time efficiency being an overlooked factor that impacts practicality of moderation solutions. Their results indicate the JudgeRail approach has improved latency over baselines (46% to 55% less time than LlamaGuard and ShieldGemma).

**Weaknesses:**

- It seems like the results are actually quite mixed for JudgeRail (table 2), and while there is lower latency, in practice I don't believe the difference would make a significant enough difference to offset the lower performance. Accuracy is more of a concern. For HateXplain, JudgeRail does better but the difference is marginal and doesn't appear to be statistically significant.

- The experimentation lacks rigor. There were several baselines highlighted in the paper, but these were absent from comparison until the authors' rebuttal. Given the limited time for experimentation, the validity of these results is questionable. The paper would benefit from statistical significance testing. It also seems like the proposed approach would be highly brittle to the number and selection of in-context examples. While the GPT-4 labeling comparison is interesting, I'm not sure why the authors chose to denote the models' agreement with GPT-4 as the "fixed" results. Are the authors convinced these were errors in the original human-annotated dataset? If so, this needs better analysis than re-labeling with GPT-4, which has its own biases and may miss nuances that human annotators observe like dialectical variation or sarcasm. I do not find it very surprising that open LLMs would produce more similar results to GPT-4 than human annotators.

- I do not agree with the authors that computational restrictions on training content moderation models is a major problem in content moderation. Training a 7b parameter model with the corpora sizes mentioned is feasible on smaller GPUs than an A100, and while this will be more time-consuming, even in industry there is rarely the need for continuous retraining. Additionally, there are many cloud computing resources available to researchers, sometimes at no cost. I think financial/psychological cost of label verification / human annotation for expanding existing corpora is a stronger argument to avoid retraining.

Overall, I am not entirely convinced by either the results or motivation of the paper, and cannot recommend acceptance yet. However, I do think there are many positive aspects of the paper and the JudgeRail framework is a very solid, interesting idea. Any NLP/AI venue would be suitable for the paper, so I suggest the authors take a bit more time to refine it and resubmit.

**Additional Comments On Reviewer Discussion:**

The reviewers agreed about the importance and timeliness of the paper's focus on content moderation. They also highlighted the discussion of latency. The main concern of the reviewers was absent baselines. There are many new results in the rebuttal that address the reviewers' questions about baselines. I believe the inclusion of these results will significantly improve the paper and strengthen the authors' arguments about the effectiveness of their approach.

I also believe the authors satisfactorily addressed in their rebuttal the comparisons to Jury Learning and approaches requiring ensembles of persona-driven models, however I agree with reviewer MwTd that the technical contribution of the paper could be improved.

---

### Decision · Program_Chairs · 2025-01-22

Reject